# Exploring Prime-Boost Vaccination Regimens with Different H1N1 Swine Influenza A Virus Strains and Vaccine Platforms

**DOI:** 10.3390/vaccines10111826

**Published:** 2022-10-29

**Authors:** Anna Parys, Elien Vandoorn, Koen Chiers, Katharina Passvogel, Walter Fuchs, Thomas C. Mettenleiter, Kristien Van Reeth

**Affiliations:** 1Laboratory of Virology, Faculty of Veterinary Medicine, Ghent University, 9820 Merelbeke, Belgium; 2Laboratory of Veterinary Pathology, Faculty of Veterinary Medicine, Ghent University, 9820 Merelbeke, Belgium; 3Institute of Molecular Virology and Cell Biology, Friedrich-Loeffler-Institut, Federal Research Institute for Animal Health, 17493 Greifswald, Germany

**Keywords:** swine influenza A virus, H1N1, vaccination, heterologous prime-boost, vectored vaccine, attenuated PrV strain Bartha, antibody cross-reactivity, ELISPOT, cross-protection

## Abstract

In a previous vaccination study in pigs, heterologous prime-boost vaccination with whole-inactivated H1N1 virus vaccines (WIV) induced superior antibody responses and protection compared to homologous prime-boost vaccination. However, no pan-H1 antibody response was induced. Therefore, to stimulate both local and systemic immune responses, we first vaccinated pigs intranasally with a pseudorabies vector vaccine expressing the pH1N1 hemagglutinin (prvCA09) followed by a homologous or heterologous WIV booster vaccine. Homologous and heterologous WIV–WIV vaccinated groups and mock-vaccinated or prvCA09 single-vaccinated pigs served as control groups. Five weeks after the second vaccination, pigs were challenged with a homologous pH1N1 or one of two heterologous H1N2 swine influenza A virus strains. A single prvCA09 vaccination resulted in complete protection against homologous challenge, and vector–WIV vaccinated groups were significantly better protected against heterologous challenge compared to the challenge control group or WIV–WIV vaccinated groups. Furthermore, vector–WIV vaccination resulted in broader hemagglutination inhibition antibody responses compared to WIV–WIV vaccination and higher numbers of antibody-secreting cells in peripheral blood, draining lymph nodes and nasal mucosa. However, even though vector–WIV vaccination induced stronger antibody responses and protection, we still failed to induce a pan-H1 antibody response.

## 1. Introduction

Three swine influenza A virus (swIAV) subtypes are circulating in swine populations worldwide: H1N1, H1N2, and H3N2. The virus strains within these subtypes differ between regions due to the geographic segregation of pig populations and multiple influenza virus introductions from either humans or birds. Furthermore, the hemagglutinin (HA) and neuraminidase (NA) surface proteins can undergo rapid evolutionary changes by mutations, called antigenic drift, and by the exchange of genome segments, called reassortment. Based on their genetic diversity, H1 swIAVs are differentiated into three major swIAV lineages, which are further subdivided into clades [1]. (i) The classical swine lineage includes descendants of the 1918 human H1N1 pandemic (1A), (ii) the human seasonal lineage derives from IAVs once circulating in humans [2,3,4] (1B), and (iii) the Eurasian avian lineage originates from an avian IAV introduced into the European swine population around 1979 (1C) [5]. In 2009, a novel H1N1 IAV (pH1N1) emerged and caused a global pandemic. The virus derived its HA gene from the North American classical swine lineage and its NA gene from a Eurasian avian H1N1 swIAV, and it circulates both in humans and swine [6]. In humans, pH1N1 viruses replaced human seasonal H1N1 IAVs and continued circulating seasonally worldwide. However, in pigs, pH1N1 viruses circulate together with other swIAV lineages and clades. Unlike these other swIAV clades, pH1N1 viruses are circulating in pigs globally, which is largely due to repeated reintroductions from humans to swine [7,8,9].

The ideal swIAV vaccine should be safe and able to induce both local and systemic immune responses. Whole inactivated swIAV vaccines (WIVs) are considered safe but mainly rely on the induction of serum antibodies against the highly variable HA, which can be hampered in young piglets due to the presence of maternally derived antibodies [10]. In contrast, live attenuated influenza vaccines (LAIVs) can induce both local and systemic immune responses, but they are considered less safe because of the risk of reassortment with endemic swIAVs in the field [11]. Viral vector vaccines can overcome these drawbacks and have been demonstrated to induce a strong cell-mediated and mucosal immune response against multiple viral infections [12,13,14,15,16,17,18]. The pseudorabies virus (PrV) strain Bartha has been used as a commercial vaccine against Aujeszky’s disease and as a vector to express viral antigens such as the HA of the prototype pH1N1 IAV strain A/California/04/2009 (prvCA09). Complete protection against homologous IAV challenge was obtained after a single intranasal vaccination with HA-expressing PrV-BaMI-synHI (in this study renamed as prvCA09) [19], whereas two administrations of matched WIV were required to do so [10,20,21]. Recently, we have demonstrated that prime-boost vaccination with antigenically distinct, monovalent H1N1 WIVs can broaden the immune response and improve protection against heterologous H1N1 IAVs. However, this strategy could not induce a pan-H1 antibody response [22]. We aimed to further improve the heterologous prime-boost vaccination strategy by using a live viral vector, prvCA09, as a priming vaccine, to stimulate both local and systemic immune responses. We hypothesized that prvCA09 vaccination followed by a homologous or heterologous WIV vaccination results in a broader immune response and protection than two doses of matched WIV. For this purpose, we determined the serum antibody response against the vaccine strains and antigenically distinct H1 IAVs. Furthermore, we examined local and systemic virus-specific antibody-secreting cells (ASCs) and evaluated protection against homologous and heterologous H1 swIAV challenge.

## 2. Materials and Methods

### 2.1. Viruses and Vaccines

Two types of vaccines were included. (i) Adjuvanted inactivated monovalent WIVs based on three different H1N1 IAVs: a pH1N1 IAV A/California/04/2009 (CA09) (1A.3.3.2), an avian-like swIAV A/swine/Gent/28/2010 (G10) (1C.2.2), and a European human-like swIAV A/swine/Cotes d’Armor/0046/2008 (ARM08) (1B.1.2.3). CA09 was obtained from the Centers for Disease Control and Prevention (CDC) and ARM08 from the Ploufragan-Plouzané-Niort Laboratory for the French Agency for Food, Environmental and Occupational Health & Safety (ANSES, Ploufragan, France). (ii) The vector vaccine PrV-BaMI-synH1 (prvCa09), based on the PrV vaccine strain Bartha, which was used as a platform for the HA of pH1N1 IAV A/Regensburg/D6/2009 (1A.3.3.2) [19,23]. For A/Regensburg/D6/2009, we also used the abbreviation CA09 since both pH1N1 IAVs are genetically and antigenically almost identical, with 98% identity in their HA1 amino acid (aa) sequences. The Bartha strain without HA insert and phosphate-buffered saline (PBS) with 20% commercial oil-in-water adjuvant (Emulsigen, MVP Laboratories, NE, USA) served as mock vaccines for prvCA09 and WIV, respectively. The WIVs were prepared and adjuvanted with Emulsigen as described previously [24,25]. Each vaccine dose contained 256 hemagglutinating units (HAU)/2 mL. The vaccine was administered intramuscularly via deep injection in the neck muscles. The prvCA09 vaccine was used at a dose of 10^7,3^ TCID_50_/3 mL and administered intranasally.

Three different challenge viruses were included: homologous challenge virus CA09 and 2 heterologous challenge viruses, European human-like swIAV A/swine/Gent/26/2012 (G12) (1B.1.2.1) and North American human-like swIAV A/swine/Illinois/A01047020/2010 (IL10) (1B.2.2.2). The latter was obtained from the U.S. Department of Agriculture (USDA) swine influenza repository held at the National Veterinary Service Laboratories.

A panel of 24 antigenically different H1 IAVs was used for hemagglutination inhibition (HI) assays. This virus panel included vaccine and challenge viruses, H1 swIAVs from Europe and North America, and human seasonal IAVs spanning 70 years of antigenic drift (Figure 1). We obtained European swIAV A/swine/Italy/7704/2001 and corresponding swine serum from the Istituto Zooprofilattico Sperimentale della Lombardia e dell’Emilia Romagna “Bruno Ubertini” (IZSLER, Brescia, Italy). The North American swIAVs and corresponding swine sera were obtained from the USDA or the University of Wisconsin–Madison. Human seasonal IAVs and corresponding ferret sera were obtained from the Francis Crick Institute or the World Health Organization (WHO).

To determine the genetic and antigenic relationship between the vaccine and challenge strains and the virus strains used for serology, protein sequences were aligned with ClustalW, and the P sequence values and P all antigenic site values for the HA1 were determined [26,27,28]. The P values are distance measures to evaluate the match between the vaccine strains and other strains. They can be determined for the entire HA1 or for aa in antigenic sites only. The greater the P value, the greater the genetic/antigenic difference. In humans, a P all antigenic site value ≤0.442 has been correlated with vaccine effectiveness greater than zero [27], but no such information is available for swine. The P sequence value is defined as the number of aa substitutions in the HA1 divided by the total number of aa. The P all antigenic site value is the number of aa substitutions in the antigenic sites divided by the total number of aa in the antigenic sites. Based on the protein sequences, a maximum likelihood tree was constructed (Figure 1) using MEGA7 based on the Jones–Taylor–Thornton substitution model with a gamma (Γ) distribution among site rate. Branch length is proportional to genetic distance.

### 2.2. Experimental Design

The study consisted of two separate experiments. In total, 122 5-week-old pigs from an influenza virus-free farm were used. Pigs were housed in a biosafety level 2 (BSL2) containment animal facility. At arrival, serum samples of all individual pigs were confirmed to be seronegative in a competitive anti-IAV nucleoprotein enzyme-linked immunosorbent assay (ID-VET, France) and in HI assays against H1N1, H1N2, and H3N2 influenza virus strains that are representative for swIAVs currently circulating in Europe. The first experiment included 98 pigs to evaluate the serum antibody response and protection against challenge when pigs are vaccinated with wivCA09 or prvCA09 as a priming vaccine and boosted with wivCA09, wivG10, or wivARM08. The experimental design is summarized in Table 1. Pigs were vaccinated twice at a 4-week interval. Two control groups received PrV Bartha without HA insert as a mock vaccine as primer and PBS with adjuvant as a booster vaccine. (i) One control group was a control for histopathological examinations. (ii) The other control group was challenged and served as a challenge control group. A third control group first received prvCA09, followed by PBS with adjuvant. Three groups were vaccinated with WIV–WIV in a homologous (wivCA09–wivCA09) or heterologous (wivCA09–wivG10 or wivCA09–wivARM08) prime-boost regimen. Another three groups were vaccinated with vector–WIV in a homologous (prvCA09–wivCA09) or heterologous (prvCA09–wivG10 or prvCA09–wivARM08) prime-boost regimen.

Five weeks after the second vaccination, pigs were challenged intranasally with 10^7^ TCID_50_/5 mL of the homologous IAV CA09 or with one out of the two heterologous challenge viruses, G12 and IL10. Nasal swabs were collected on the day of challenge and three days post-challenge. Three days post-challenge, pigs were humanely euthanized with a lethal dose of pentobarbital followed by exsanguination. The trachea and lungs were removed to calculate the percentage of gross lung lesions, and samples of the distal part of the trachea and the right cardiac lung lobe were used for microscopic histopathological examination. To evaluate virus titers in the respiratory tract, 20% of tissue homogenates of the right and left lung, trachea, and nasal mucosa were examined. Blood samples for serum were collected at arrival, 14 days post each vaccination and on the day of challenge.

In the second experiment, 24 pigs were used. In this experiment we aimed to compare the local and systemic ASC responses following vaccination with wivCA09–wivARM08 or prvCA09–wivARM08. These vaccination regimens were selected because they performed best in the first experiment. Mock-vaccinated pigs were included as controls. ASCs were isolated from peripheral blood (PBMC), tracheobronchial lymph nodes (TBLN), lymph nodes of the head (LNH), including retropharyngeal, parotid, and superficial cervical lymph nodes, and the nasal mucosa (NMC) as described previously [25]. Pigs were sampled 14 days post the first vaccination (week 2), at the time of the second vaccination (week 4), and seven days post the second vaccination (week 5). In previous studies, ASC responses in these samples have been shown to peak at these time points [25,29]. At each time point, we euthanized three pigs from both vaccinated groups as well as two mock-vaccinated control pigs. Mononuclear cells from the above-mentioned samples were examined for IgG and IgA ASCs against virus strains used for immunization (CA09, G10, and ARM08) and challenge (CA09, G12, and IL10) in an enzyme-linked immunospot assay (ELISPOT).

### 2.3. Hemagglutination Inhibition Assay

Serum samples were examined in HI assays to determine antibody titers against the three virus strains used for vaccination. Serum samples collected on the day of challenge were tested against a broader panel of 24 H1 virus strains. The HI test was performed according to standard procedures [30], and the starting dilution was 1:10. HI titers represent the reciprocal of the highest serum dilution that inhibited hemagglutination of 4 hemagglutinating units (HAU) of virus.

Sera were inactivated at 56 °C for 30 min to eliminate non-specific HI factors and subsequently treated with receptor-destroying enzyme at 37 °C overnight. Afterward, samples were absorbed into turkey erythrocytes to remove agglutinating factors present in swine serum. Samples were tested in 2-fold dilutions and incubated with 4 HAU of virus. After incubation at room temperature for 1 h, 0.5% of turkey red blood cells were added, and hemagglutination patterns were read after 1 h.

### 2.4. Virus Titration and Lesion Scores

Tissue samples were collected from the apical, cardiac, and diaphragmatic lung lobes of both the left and right lung half of each pig. The three lung samples of each half were pooled. Virus titers were evaluated in nasal swabs and in 20% homogenates of the nasal mucosa, trachea, and right and left lung as previously described [31]. Ten-fold serial dilutions of samples were inoculated onto Madin–Darby Canine Kidney (MDCK) cells. The cells were observed daily for seven days for development of cytopathic effect (CPE). Virus titers were calculated by the method of Reed and Muench and expressed as log_10_TCID_50_ per g tissue, or per 100 mg nasal secrete [32].

At necropsy, we determined the percentage of macroscopic pneumonia by assessing purple-red lesions typical of swIAV infection. The percentage of affected lung surface area was calculated as previously described [33,34]. Tissue samples from the trachea and right cardiac lung were fixed in 4% buffered formalin and processed for histopathological examination as previously described [33]. Microscopic lesions in the lung and trachea were evaluated by a veterinary pathologist blinded to treatment groups and scored according to previously described parameters [33].

### 2.5. ELISPOT Analysis

Tissue collection, processing, and mononuclear cell isolation were conducted as described previously [25]. The mononuclear cells were isolated and resuspended in complete RMPI medium (RMPI 1640-GlutaMax medium (Life Technologies, Paisley, UK) with 1% sodium pyruvate, 1% nonessential aa, 10% fetal calf serum, 1% penicillin-streptomycin, and 0.5% gentamicin).

The ASC IgG and IgA ELISPOT assays were conducted as described previously [25]. Briefly, 96-well ELISPOT filter plates (Merck Millipore, Molsheim, France. Cat. No. MAIPS4510) were incubated with PBS or 200 HAU purified live virus per well at 4 °C for 16 h. Afterward, plates were washed with PBS and blocked with complete RPMI medium. Mononuclear cells were dispensed in duplicates into the wells and incubated for 18 h at 37 °C and 5% CO_2_. Plates were washed and incubated for 2 h at 37 °C with monoclonal biotinylated mouse anti-porcine IgG or monoclonal mouse anti-porcine IgA to capture the antibodies secreted by the ASCs. After washing, streptavidin horseradish peroxidase (HRP) (3310-9-1000, MABTECH Inc., Cincinnati, OH, USA) was used for IgG and HRP-labeled goat anti-mouse immunoglobulins (Dako, Glostrup, Denmark, P0447) for IgA. Following 1 h incubation at 37 °C, IgG ASCs were visualized with TMB substrate for ELISPOT (3651-10, MABTECH Inc., Cincinnati, OH, USA) and IgA ASCs with TMB Liquid Substrate System for Membranes (Sigma-Aldrich, Saint Louis, MO, USA). Blue spots, corresponding to activated ASC, were counted.

### 2.6. Statistical Analysis

Log2-transformed antibody titers against the vaccine strains, lesion scores, and log10-transformed virus titers were analyzed using the Kruskall–Wallis test followed by Dunn’s test. Numbers of ASCs were compared between groups using the Mann–Whitney U test. A *p* value of ≤0.05 was considered significant. If only three observations per group were available, which was the case for ASC numbers, *p* values of ≤0.1 were considered significant. Antibody titers below the detection limit (<10) were assigned a value of 5, and virus titers below the detection limit (1.7 log_10_ TCID_50_) were assigned a value of 0.85 log_10_ TCID_50_.

## 3. Results

### 3.1. HI Antibody Responses against Vaccine Strains

Three antigenically and genetically distant vaccine strains were used for priming and booster vaccination: CA09, G10, and ARM08. Pairwise comparisons of the HA1 aa sequences of CA09 with G10 and ARM08 resulted in P sequence values of 0.27 and 0.28, and P all antigenic site values of 0.32 and 0.54 (Figure 1). Antibodies against the vaccine strains were examined by the HI assay (Figure 2). IAV antibodies were undetectable in all pigs prior to vaccination, and the challenge control group remained seronegative throughout the experiment.

Fourteen days after the first vaccination (week 2), HI antibody titers were only observed against CA09 and were higher in prvCA09 vaccinated pigs than in wivCA09 vaccinated pigs (*p* ≤ 0.01). After the second vaccination (week 6), HI antibody titers peaked against all three vaccine strains. Compared to group prvCA09–PBS, higher HI antibody titers against CA09 were observed in both wivCA09 boosted groups (*p* < 0.001). Furthermore, the wivCA09 booster vaccine induced HI antibody titers against G10 and ARM08. HI antibody titers against G10 were higher in groups that received a wivG10 booster vaccine than in groups that received a wivARM08 booster (*p* ≤ 0.01). Likewise, HI antibody titers against ARM08 were higher in groups with a wivARM08 booster (*p* ≤ 0.004). Three weeks later, on the day of the challenge (week 9), HI antibody titers followed a similar pattern as in week 6, but titers were up to 7-fold lower.

In summary, after the first vaccination, prvCA09 vaccinated pigs had significantly higher titers against CA09 than wivCA09 vaccinated pigs. However, after the booster vaccination, there was no difference between WIV–WIV and vector–WIV vaccinated groups. After a wivG10 or a wivARM08 boost, antibodies were induced against the primary and booster vaccine strain. On the other hand, a wivCA09 booster induced antibodies against all three vaccine strains.

### 3.2. HI Antibody Responses against Antigenically Diverse H1 IAVs of Swine and Humans

To evaluate the breadth of antibody responses, we selected a panel of 24 antigenically different H1 IAVs that included vaccine and challenge viruses, H1 swIAVs from Europe and North America, and human seasonal IAVs. Both heterologous challenge viruses, G12 and IL10, were genetically and antigenically distant from the vaccine strains with P sequence values between 0.10 and 0.31 and P all antigenic site values between 0.18 and 0.62. The P sequence values between the vaccine strains and H1 swIAVs from Europe and North America varied between 0.05 and 0.31 and P all antigenic site values between 0.10 and 0.64. With human seasonal IAVs, vaccine strains shared P sequence values between 0.09 and 0.28 and P all antigenic site values between 0.18 and 0.56 (Figure 1).

Pooled sera from each group, collected on the day of the challenge, were examined for HI antibody titers against this H1 IAV panel. To compare the HI antibody responses between groups, we implemented a scoring system (0–4.5, Figure 3) based on accepted seroprotective thresholds for seasonal influenza in vaccinated humans (HI titer ≥ 40 = HI titer score ≥ 2) and on data from vaccination challenge studies with influenza-naïve pigs (HI titer ≥ 160 = HI titer score ≥ 3). For each group, we calculated the cumulative HI titer scores (titer score; max. 120) and converted them to percentages. Additionally, we calculated the percentage of virus strains against which an HI titer score ≥ 2 was reached (cross-reactivity).

No IAV antibodies were detected in the mock-vaccinated challenge control group. The prvCA09–PBS group only had HI titer scores ≥ 2 against virus strains from clade 1A.3.3, with low titer-(8%) and cross-reactivity scores (13%). After two vaccine administrations, titer- and cross-reactivity scores were up to three times higher. Homologous prime-boost vaccination wivCA09–wivCA09 induced HI titer scores ≥ 2 against swIAVs of the same lineage as the vaccine strain (1A) and against virus strains from the Eurasian avian lineage (1C) and European human-like H1 viruses (1B.1). This resulted in a titer- and cross-reactivity score of 24% and 33%, respectively. Heterologous prime-boost vaccination wivCA09–wivG10 and wivCA09–wivARM08 induced similar scores, which at best resembled those of the homologous prime-boost group wivCA09–wivCA09. The serological profiles of the vector–WIV vaccinated groups were similar to those of the WIV–WIV counterparts, with a maximum score difference of 4%. Only group prvCA09–wivARM08 reached a cross-reactivity score of 38% and outscored all other vaccinated groups.

HI antibody titers were invariably highest against the first administered vaccine strain CA09. On the other hand, no or low HI antibody titer scores were observed against North American human-like H1 swIAVs (1B.2), including challenge virus IL10 or human seasonal H1 IAVs. Against challenge virus G12, the seroprotective threshold (HI titer score ≥ 2) was only reached by the groups that received a wivARM08 booster vaccine. Overall, similar titer and cross-reactivity scores were observed in all vaccinated groups, with the highest cross-reactivity score in group prvCA09–wivARM08.

### 3.3. Protection against Homologous and Heterologous Virus Challenge

Five weeks after the last vaccination, pigs were challenged with the homologous challenge virus CA09 or one out of the two heterologous challenge viruses, G12 or IL10. Three days post-challenge, pigs were euthanized to examine virus titers in the respiratory tract (Figure 4) and macroscopic and microscopic lesions (Table 2).

All three challenge viruses replicated to high titers in all samples of the mock-vaccinated challenge control pigs (Figure 4). After a homologous challenge with CA09, a single prvCA09 vaccination offered sterile protection. However, none of the vaccinated groups was fully protected against heterologous challenge. G12 virus titers were similar to those of the challenge control group in groups wivCA09–wivCA09 and wivCA09–wivG10. On the other hand, mean virus titers in the lungs of group wivCA09–wivARM08 were reduced compared to the challenge control group (*p* ≤ 0.02). Furthermore, mean viral titers in the nasal mucosa were reduced compared to group wivCA09–wivG10 (*p* = 0.04). In the vector–WIV vaccinated groups, including group prvCA09–PBS, G12 virus titers were reduced in 1 or more samples compared to the challenge control group or group wivCA09–wivG10 (*p* ≤ 0.05). None of the vaccination regimens offered virological protection against challenge with IL10. 

The four unvaccinated, unchallenged control pigs had no lesions in the respiratory tract, except for one pig with peribronchiolar lymphocytic cuffing in the lung (results not shown). Challenge with CA09 caused only mild macroscopic and microscopic lung lesions and no tracheal lesions in both the challenge control group and vaccinated pigs (Table 2). Challenge with G12 mainly caused macroscopic and microscopic lesions in the challenge control group and the groups that received a wivCA09 booster vaccine. Despite the high viral titers after IL10 challenge, macroscopic and microscopic lesions were mild in pigs of all groups.

In conclusion, a single prvCA09 vaccination offered sterile protection against homologous challenge. Protection against challenge with G12 improved when pigs were primed with prvCA09 and/or boosted with wivARM08. No protection was induced against IL10. Lesions of the respiratory tract were minimal, and there were no significant differences between groups.

### 3.4. Antibody-Secreting Cell Responses against Vaccine and Challenge Viruses

To examine the local and systemic virus-specific ASC responses following vaccination, we performed a second experiment with the two vaccination regimens that had shown the highest cross-reactivity score in the first experiment and/or the best protection against heterologous challenge with G12: wivCA09–wivARM08 and prvCA09–wivARM08 (Figure 5). Mock-vaccinated pigs were included as controls and tested negative in the ELISPOT assays throughout the experiment.

Fourteen days after the first vaccination (week 2), ASC responses were mainly present against the homologous vaccine strain CA09. Group wivCA09–wivARM08 mainly had IgG ASCs in TBLN and LNH, while group prvCA09–wivARM08 had higher numbers of IgA ASCs in NMC than group wivCA09–wivARM08 (*p* = 0.1). At the time of the second vaccination (week 4), group wivCA09–wivARM08 had low levels of ASCs in all samples. On the other hand, its vector–WIV counterpart had high numbers of IgG ASCs in TBLN and IgG and IgA ASCs in NMC (*p* = 0.1).

Seven days after the second vaccination (week 5), IgG ASCs against CA09 had increased up to 16-fold in PBMCs and LNH in both groups, and group prvCA09–wivARM08 had higher numbers of IgG ASCs in PBMCs (*p* = 0.1). Higher numbers of IgG and IgA ASCs were also observed in NMC (*p* = 0.1). At this time point, the effect of the booster vaccination was evident in both groups, and IgG and IgA ASCs against ARM08, and to a lesser extent against G12, were elevated in all tissues. Priming with prvCA09 had a beneficial effect on anti-ARM08 IgG and IgA ASCs in PBMCs and IgG ASCs in NMC (*p* = 0.1). Responses against G10 and IL10 were minimal.

Throughout the experiment, group prvCA09–wivARM08 had higher ASC responses in NMC than group wivCA09–wivARM08. For group wivCA09–wivARM08, the draining lymph nodes were the predominant location for IgG ASCs and NMC for IgA ASCs. For group prvCA09–wivARM08, NMC seemed to be the primary source for both antibody isotypes. Furthermore, the numbers of IgG ASCs were up to 14-times higher than IgA ASCs in both groups, except in NMC, where the proportion between both isotypes was more balanced.

## 4. Discussion

This study builds on a previous H1 heterologous prime-boost vaccination study [22], in which we failed to induce a pan-H1 neutralizing antibody response and protection by heterologous prime-boost vaccination with antigenically distinct, monovalent H1N1 WIVs. In the present study, we tried to improve the efficacy of heterologous prime-boost vaccination by intranasal priming with a prvCA09 vector instead of an intramuscular administration of WIV. This way, we aimed to enhance both local and systemic immune responses and thus improve the overall immune response and protection. Our results show that a single prvCA09 vaccination induces higher HI antibody titers against CA09 than a single wivCA09 vaccination. Furthermore, a single vector vaccination resulted in complete protection against homologous challenge, whereas, in influenza naïve pigs, at least two WIV vaccinations are required to induce complete protection against homologous challenge [10]. After the booster vaccination, similar HI antibody responses were observed with WIV–WIV combinations and their vector–WIV counterpart, yet the highest cross-reactivity score was induced by prvCA09–wivARM08. Overall, ASC responses were significantly higher in group prvCA09–wivARM08 than in group wivCA09–wivARM08, with NMC as a predominant location for IgG and IgA ASCs in the former group. Although we partly succeeded in our objective and the vector vaccination improved the immune response and protection, we failed to induce a pan-H1 response and complete protection against heterologous challenge.

The prvCA09 vaccination elicited high numbers of anti-CA09 IgA and IgG ASCs in NMC and IgG ASCs in TBLN. However, IgA ASC responses were up to 25-fold and IgG responses up to 5-fold lower than those detected after intranasal infection of pigs with a classical H1N1 swIAV (1A.1-like) in a study by Larsen et al. [29]. On the other hand, the kinetics of ASC responses following prvCA09 vaccination were similar to those after infection [25,29], with a delayed IgG ASC response (28 dpv1/0 dpv2) compared to the IgA ASC response (14 dpv1). There are multiple possible explanations for the lower number of ASCs following prvCA09 vaccination as compared to an influenza virus infection. First, the prvCA09 vaccine only includes the HA influenza protein. Though anti-HA antibodies are virus-neutralizing, ASCs against other influenza proteins, which are not included in the prvCA09 vaccine, may contribute to the immune response and the total number of ASCs following infection [35,36]. Secondly, Klingbeil and colleagues [19] observed that, despite abundant expression of HA in cells infected with prvCA09, HA was not included in the prv virions upon viral release. This may limit the exposure of the immune system to the HA antigen in prvCA09 vaccinated pigs and result in lower numbers of ASCs. Lastly, both influenza and PrV use the nasal mucosa as a portal of entry, but the virus replication cycle and pathobiology are different, which may also account for differences in the immune response. Following the wivARM08 booster vaccination, ASCs were mainly detected in the draining lymph nodes (TBLN and LNH) in wivCA09–wivARM08 vaccinated pigs, whereas all four samples examined had high numbers of ASCs in prvCA09–wivARM08 vaccinated pigs. The detection of high numbers of ASCs in all four examined tissues is consistent with findings by Jegeskanda and colleagues [37]. These researchers demonstrated that a priming mucosal vaccination induces ASCs at the site of vaccination, which can be recalled at numerous distinct sites following a WIV booster vaccination.

The prvCA09–wivG10 vaccinated pigs were better protected against heterologous challenge with G12 than with IL10. Remarkably, both heterologous challenge strains showed a similar genetic distance from the CA09 and G10 vaccine strains. However, the exact aa differences differ between G12 and IL10. It is well-known that the nature of these aa differences is of greater importance than their number [38]. Therefore we speculate that the difference in protection against G12 and IL10 in prvCA09–wivG10 vaccinated pigs relates to qualitative aa differences between vaccine and challenge viruses rather than quantitative differences. Furthermore, protection against heterologous G12 challenge was generally better after priming with prvCA09 than after priming with wivCA09, and the difference was most prominent with the CA09-G10 combination. However, pre-challenge HI antibody titers against G12 did not differ between vector–WIV and WIV–WIV vaccinated groups. This supports the notion that mucosal vaccination induces a more multifaceted and complex immune response than WIV vaccination [12,39]. Indeed, vector–WIV vaccination induced higher numbers of local and systemic ASCs than WIV–WIV vaccination. So, following mucosal vaccination, one cannot rely on HI antibody responses alone to estimate protection against a given influenza virus [40]. One limitation is that we only partly evaluated the mucosal and systemic antibody responses and not T-cell mediated responses, which may also contribute to the induced cross-protection. Nevertheless, our data demonstrate the benefits of using both traditional (WIV) and novel (vector) immunization platforms in a heterologous prime-boost vaccination strategy [41,42].

Overall, vector–WIV vaccination resulted in broader antibody responses and protection than WIV–WIV vaccination. Furthermore, vector–WIV vaccination might have additional advantages. First, as illustrated by Vincent and colleagues [35], mucosal vaccination is less likely to interfere with colostrum-borne maternal antibodies. Several vector vaccines can be administered intranasally, and maternally derived antibodies are predominantly of the IgG isotype and unlikely to infiltrate the nasal mucosa of the upper respiratory tract. Secondly, vector vaccines can elicit both local and systemic immune responses, unlike WIVs. Lastly, vector vaccines are considered safer than LAIVs. Moreover, domestic pigs in most European countries and in North American countries are officially free of PrV [43]. Thus, there is no risk of interference with preexisting immunity or maternally-derived antibodies with active immunization by the vector vaccine [44,45,46]. Our study confirms that mucosal vaccination primes influenza-naïve pigs for a more robust mucosal immune response following intramuscular vaccination with WIVs [37]. Though vector–WIV vaccination still failed to induce a pan-H1 immune response or complete protection against heterologous challenge, our study highlights the value of vector vaccines for use in heterologous prime-boost vaccination regimens.

## Figures and Tables

**Figure 1 vaccines-10-01826-f001:**
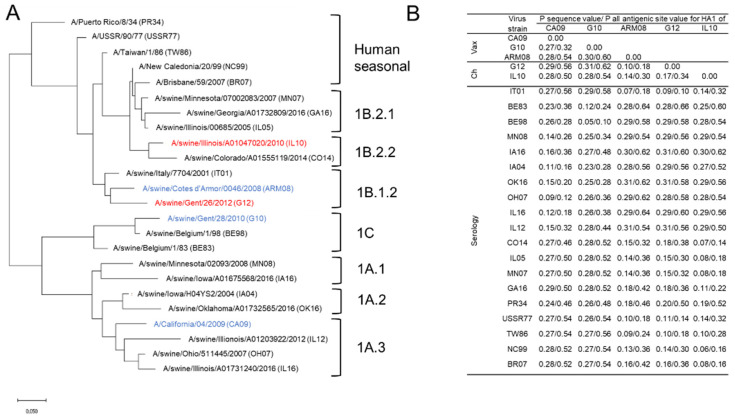
Genetic relationship between the influenza A virus strains included in the study. Maximum likelihood phylogenetic tree based on the hemagglutinin 1 (HA1) amino acid sequences (**A**) and P sequence values and P all antigenic site values for the HA1 (**B**) were determined for the influenza A virus strains used for vaccination (Vax) (blue), challenge (Ch) (red) and hemagglutination inhibition assays (serology). The H1 lineage is indicated: 1A, Classical swine lineage, 1B Human seasonal lineage, 1C Eurasian avian lineage.

**Figure 2 vaccines-10-01826-f002:**
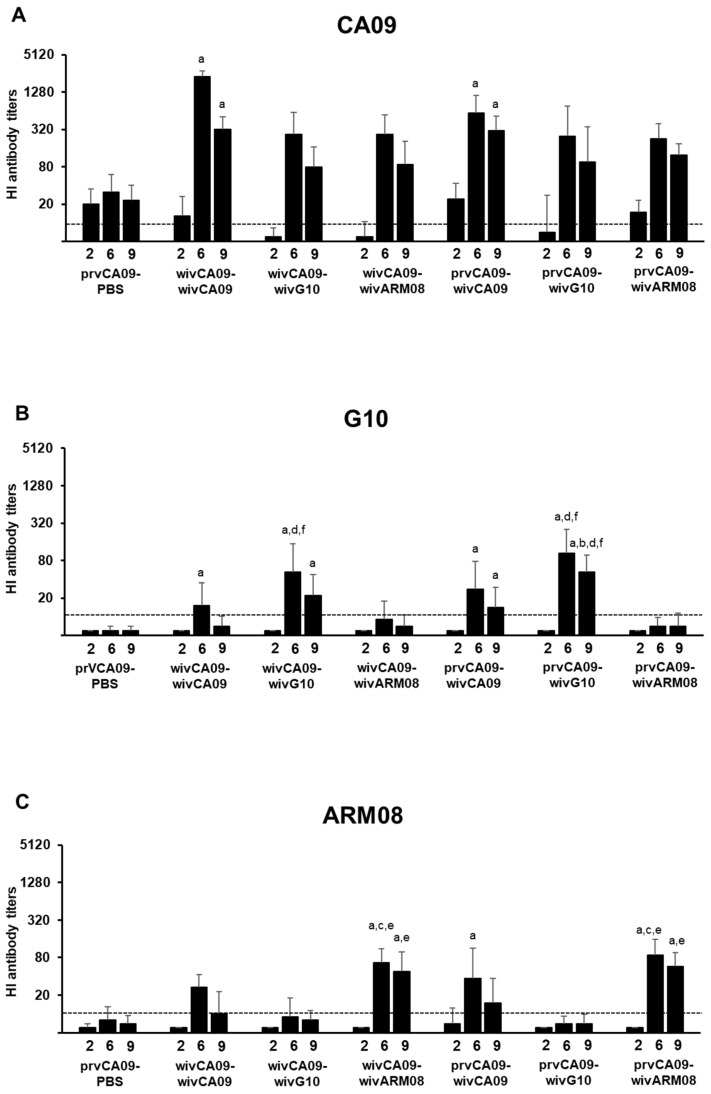
Geometric mean hemagglutination inhibition antibody titers against vaccine strains CA09 (**A**), G10 (**B**), and ARM08 (**C**). Titers were determined at 2 weeks post-vaccination 1 and 2 and at challenge (weeks 2, 6, and 9). The vaccine groups are described on the x-axis. The dotted line indicates the detection limit (10). Mock-vaccinated control pigs tested negative in all assays and were not shown. Letters indicate significant differences to prvCA09–PBS (a), wivCA09–wivCA09 (b), wivCA09–wivG10 (c), wivCA09–wivARM08 (d), prvCA09–wivG10 (e), prvCA09–wivARM08 (f) in the Kruskall–Wallis test (*p* ≤ 0.05). See Figure 1 for full names of the virus strains.

**Figure 3 vaccines-10-01826-f003:**
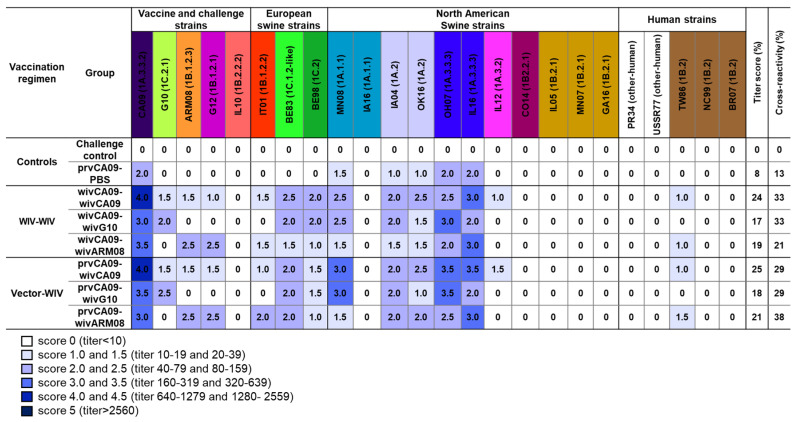
Hemagglutination inhibition (HI) titer scores against a panel of 24 H1 influenza A virus strains on the day of challenge. The virus strain colors are based on the HA clade, which is mentioned between brackets [1]. Each row represents the data generated with pooled serum from 1 vaccine group. Antibody titers were given a score from 0–4.5; the scoring system is explained below the figure. For each group, we calculated the cumulative HI titer scores (titer score; max. 120) and converted them to percentages. Additionally, we calculated the percentage of virus strains against which an HI titer score ≥ 2 was reached (cross-reactivity). See Figure 1 for full names of the virus strains.

**Figure 4 vaccines-10-01826-f004:**
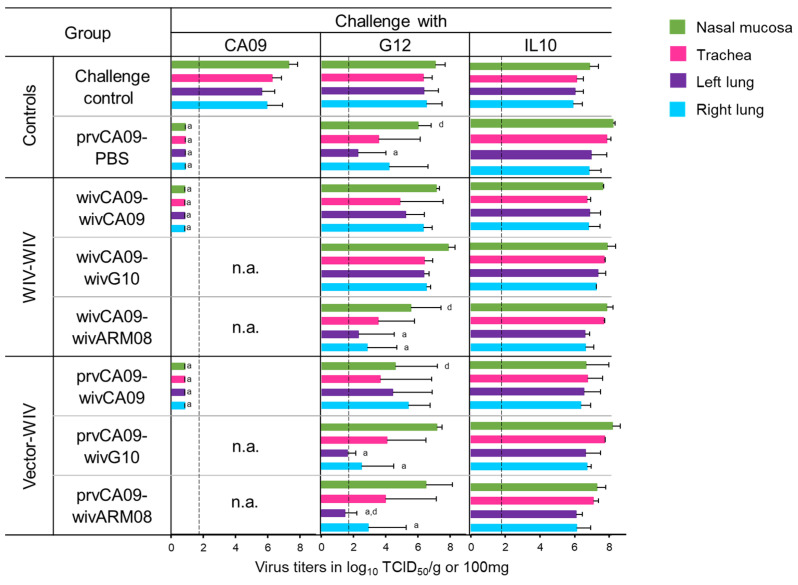
Virus titers in the respiratory tract three days post-challenge with CA09, G12, and IL10. The bars represent mean virus titers and their standard deviation. Each color represents a different tissue. Dotted lines indicate the detection limit (1.7). Virus titers in nasal swabs are expressed in log10 TCID50/100 mg nasal secrete, virus titers in the trachea and lung are expressed in log10 TCID50/g tissue. Letters indicate significant differences to challenge control (a), prvCA09–PBS (b), wivCA09–wivCA09 (c), wivCA09–wivG10 (d), wivCA09–wivARM08 (e), prvCA09–wivG10 (f), prvCA09–wivARM08 (g) in the Kruskall–Wallis test (*p* ≤ 0.05). See Figure 1 for full names of virus strains.

**Figure 5 vaccines-10-01826-f005:**
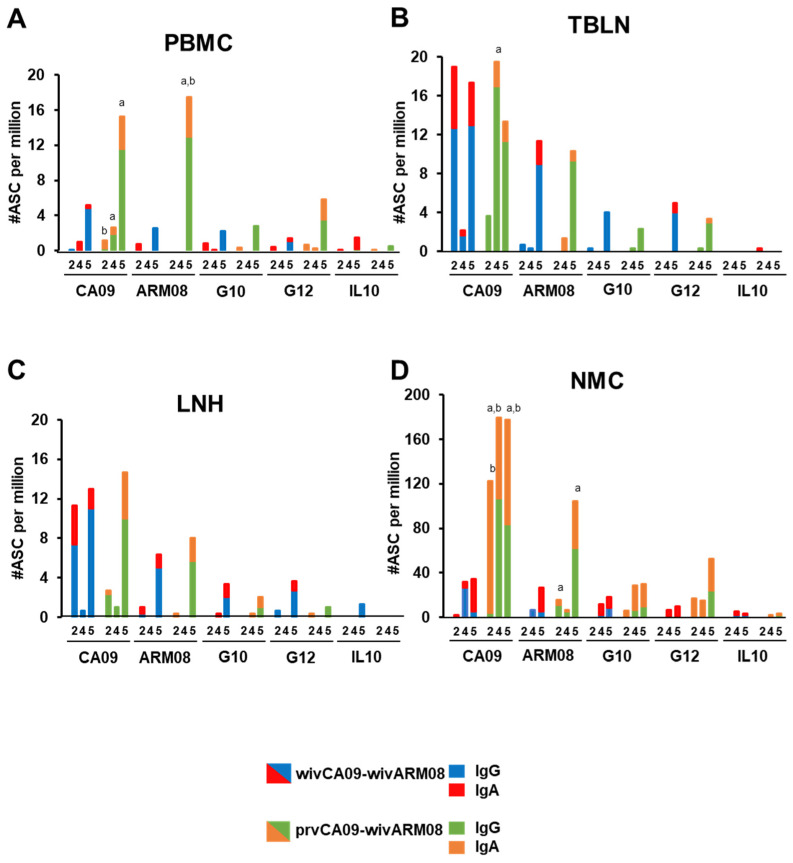
Mean numbers (#) of virus-specific IgG and IgA antibody-secreting cells (ASCs) against vaccine and challenge viruses determined by ELISPOT in peripheral blood mononuclear cells (PBMC) (**A**), tracheobronchial lymph nodes (TBLN) (**B**), lymph nodes of the head (LNH) (**C**) and nasal mucosa (NMC) (**D**) after wivCA09–wivARM08 vaccination (blue and red) or prvCA09–wivARM08 vaccination (green and orange). Responses were examined 2 and 4 weeks post-vaccination 1, and 1 week post-vaccination 2 (weeks 2, 4, and 5). Mock-vaccinated control pigs tested negative in all assays and are not shown. Letters indicate significant higher numbers of virus-specific IgG (a) or IgA (b) ASCs in group prvCA09–wivARM08 compared to group wivCA09–wivARM08 in the Mann–Whitney U test (*p* ≤ 0.1).

**Table 1 vaccines-10-01826-t001:** Experimental design of experiment 1.

		Vaccinations		No. Pigs			
Vaccination Regimen	Group	First (Week 0)	Second (Week 4)	Total	Challenge (Week 9)		
					CA09	G12	IL10
Controls	Unvaccinated unchallenged control	prv(-)	PBS	4	0	0	0
	Unvaccinated challenged control	prv(-)	PBS	26	6	10	10
	prvCA09–PBS	prvCA09	PBS	12	4	4	4
WIV–WIV	wivCA09–wivCA09	wivCA09	wivCA09	12	4	4	4
	wivCA09–wivG10	wivCA09	wivG10	8	0	4	4
	wivCA09–wivARM08	wivCA09	wivARM08	8	0	4	4
Vector–WIV	prvCA09–wivCA09	prvCA09	wivCA09	12	4	4	4
	prvCA09–wivG10	prvCA09	wivG10	8	0	4	4
	prvCA09–wivARM08	prvCA09	wivARM08	8	0	4	4

prv(-): Pseudorabies vector vaccine without HA insert; PBS: phosphate buffered saline; wiv: whole inactivated vaccine; prvCA09: Pseudorabies vector vaccine with pH1N1 HA insert; See Figure 1 for full names of IAVs.

**Table 2 vaccines-10-01826-t002:** Trachea and lung pathology after homologous and heterologous virus challenge.

		% Macroscopic Pneumonia	Microscopic Lesion Score			% Macroscopic Pneumonia	Microscopic Lesion Score			% Macroscopic Pneumonia	Microscopic Lesion Score	
Group	*n*		Lung	Trachea	*n*		Lung	Trachea	*n*		Lung	Trachea
Unvaccinated challenged control	6	0.67 (4)	1.75 (2)	0.00 (0)	10	2.40 (6)	1.30 (8)	0.00 (0)	10	0.43 (7)	0.95 (2)	0.20 (2)
prvCA09– PBS	4	0.13 (1)	0.00 (0)	0.00 (0)	4	1.50 (2)	0.25 (1)	0.25 (1)	4	0.50 (1)	1.00 (2)	0.25 (1)
wivCA09– wivCA09	4	0.00 (0)	0.25 (1)	0.00 (0)	4	3.58 (3)	0.25 (1)	0.50 (2)	4	0.28 (2)	0.50 (2)	0.25 (1)
wivCA09– wivG10	n.a.	n.a.	n.a.	n.a.	4	0.44 (1)	0.00 (0)	0.00 (0)	4	1.50 (3)	0.50 (2)	0.25 (1)
wivCA09–wivARM08	n.a.	n.a.	n.a.	n.a.	4	0.63 (1)	0.50 (2)	0.00 (0)	4	0.28 (3)	0.25 (1)	0.25 (1)
prvCA09– wivCA09	4	0.00 (0)	0.75 (2)	0.00 (0)	4	3.00 (4)	2.50 (3)	0.00 (0)	4	0.63 (3)	2.38 (2)	0.75 (3)
prvCA09– wivG10	n.a.	n.a.	n.a.	n.a.	4	0.69 (3)	0.25 (1)	0.25 (1)	4	1.36 (4)	2.50 (2)	0.00 (0)
prvCA09–wivARM08	n.a.	n.a.	n.a.	n.a.	4	0.63 (1)	0.63 (2)	0.00 (0)	4	0.46 (3)	1.00 (2)	0.25 (1)

The number between brackets indicates the number of pigs with lesions. n.a.: not applicable; prv(-): Pseudorabies vector vaccine without HA insert; PBS: phosphate buffered saline; wiv: whole inactivated vaccine; prvCA09: Pseudorabies vector vaccine with pH1N1 HA insert. See Figure 1 for full names of swIAVs. Macroscopic pneumonia: the percentage of the surface affected with pneumonia was estimated visually for each lobe, and the total percentage for the entire lung was calculated based on weighted proportions of each lobe to the total volume. Microscopic lung lesion scores are based on the severity of 3 parameters: (1) IPAW: epithelial damage in intrapulmonary airways (0–3), (2) PBLC: peribronchiolar lymphocytic cuffing (0–3), (3) Neutro’s: neutrophil exudation in bronchioles and alveoli (0–2); Microscopic tracheal lesion scores are based on the severity of epithelial damage (0–2).

## Data Availability

The data supporting the conclusions of this article are included within the article.

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
