# Peer review of "Exploring Prime-Boost Vaccination Regimens with Different H1N1 Swine Influenza A Virus Strains and Vaccine Platforms"

_vaccines, 2022, doi:10.3390/vaccines10111826_

Round 1

Reviewer 1 Report

The manuscript of Anna Parys and colleagues exploring prime-boost vaccination regimens with different H1N1 swine influenza A virus strains in pigs. They found vector-WIV vaccinated groups were better protected against heterologous challenge compared to WIV-WIV vaccinated groups. In addition, vector-WIV vaccination resulted in broader HI antibody responses compared to WIV-WIV vaccination. However, they failed to induce a pan-H1 antibody response. The comments below can better improve the manuscript. 

Major comments:

1.     The authors simply used P sequence values and P all antigenic site values in the main text comparing the viruses/vaccines used in this study. I think it would be better for the audience to understand if the authors define what these numbers mean. For example: values <0.05 is considered small difference and etc.

2.     The authors used HA units to quantify WIV vaccine dose, which could be bias due to the binding affinity of these viruses to red blood cells used. It would be great if the authors can determine the HA proteins amount in the vaccines. 

3.     Challenge viruses G12 (0.29/0.56 with Ca09, 0.31/0.62 with G10) and IL10 (0.28/0.50, 0.28/0.54) have similar P sequence values and P all antigenic site values comparing to vaccine strains Ca09 and G10. Are the authors expecting the challenge results similar for these two vaccine strains? However, in the vector-WIV group of Ca09-G10 group, the viral titers in these two viruses seems to have a big difference (Fig 4).

Minor comments:

1.     The method for virus titration is not standard based on the WHO manual of influenza surveillance (line 206). Please explain why the authors read TCID50 results at 7 days post infection for cytopathic effect instead of 3 dpi.

Reviewer 2 Report

In this manuscript by Anna Parys et al., the authors set out to study the vaccination of pigs against swine influenza, using a primo-vaccination through the nasal route with a pseudorabies vector vaccine expressing the H1 hemagglutinin of the 2009 pandemic H1N1 (prvCA09), followed by a booster with a whole inactivated virus vaccine (WIV) corresponding to the same virus (wivCA09) or to either of two distinct H1N1 swine influenza viruses.

Rationale of the study, and main findings The context is well described in the introduction, and establishes the rationale of the present study. Indeed, previous works had shown the limitations of 2-dose heterologous prime-boost vaccination using WIV, leading the authors to propose a prime-boost vaccination using a live viral vector (pseudorabies virus, prv) for the priming vaccine. This prvCA09 was administered via the nasal route in order to induce a mucosal response. The boost vaccine was administered intramuscularly and consisted of a WIV version of the same virus (wivCA09) or from either of two other H1N1 swine influenza viruses. -          The first experiment compared 9 groups of pigs that received different vaccination regimens. This allowed notably to compare two vaccination schemes (WIV+WIV versus prv+WIV). Five weeks after the last vaccination dose, pigs were challenged intranasally with 10^7 TCID50 of either of three viruses (CA09; G12 and IL10), allowing the authors to examine the protection efficiency as well as the breadth of the heterologous protection. The immune response after vaccination was assessed by measuring the H1 antibody responses, while the level of protection was assessed relying on several pathology markers (pneumonia, lesions scores). -          This first experiment showed that there was no difference between WIV-WIV and vector-WIV vaccinated groups, as regards the antibody response measured by hemagglutinin inhibition (HI) reactions. Of the three groups receiving the prv+WIV regimen, the groups receiving either the wivCA09 or the ARM08 booster had slightly better cross-reactivity scores, according to HI scores. -          This first experiment also showed that one of the two heterologous prv+WIV regimens conferred a slightly better protection against a viral challenge, while the prvCA09 or wivCA09 conferred an almost total protection against the CA09 challenge. -          The two vaccination regimens that gave the best results in the first experiments (i.e. prvCA09-wivARM08 and wivCA09-wivARM08) were compared in a second experiment, where the immune response was assessed by measuring the antibody-secreting cells (ASC) responses in four tissues: PBMC; tracheobronchial lymph nodes; lymph nodes of the head; nasal mucosa. This revealed that ASC responses were generally higher in group prvCA09-wivARM08   In spite of its few positive results, this study is well conducted and its results are well described. I have only a few remarks.   Major remarks Lines 442-43. The authors note that after prvCA09, IgA and IgG ASC responses lower than those detected after intranasal infection of pigs with a classical H1N1 swIAV. The authors propose several explanations, but they do not discuss the nature of the viral vector. Indeed, the first target of swine influenza viruses are the cells of the nasal mucosa, but I doubt that the pseudorabies virus targets the same cells. The somewhat unconvincing results obtained with the prv vector could be linked with its tissue tropism, and even to some other criteria that are inherent to the virus, such as (non exhaustive): cellular tropism; duration and outcome of the viral cycle (including or not cell death); virus-specific pathways of the innate immune response… Perhaps the authors should also emphasize the almost total protection conferred by the wivCA09 and prvCA09 against the homologous challenge. Minor remarks

Line 161. …intranasally with 10^7 (not 107)
